# Effect of Insulin Pen Training Using the Teach-Back Method on Diabetes Self-Management, Quality of Life, and HbA1c Levels in Older Patients with Type 2 Diabetes: A Quasi-Experimental Study

**DOI:** 10.3390/healthcare12181854

**Published:** 2024-09-14

**Authors:** Tahsin Barış Değer, Huri Seval Gönderen Çakmak, Banu Cihan Erdoğan, Mustafa Özgür Değer

**Affiliations:** 1Department of Gerontology, Faculty of Health Sciences, Çankırı Karatekin University, 18100 Çankırı, Türkiye; 2Department of Geriatric Care Services, Çerkeş Vocational School, Çankırı Karatekin University, 18600 Çankırı, Türkiye; sevalgonderen@hotmail.com (H.S.G.Ç.); banucihan_09@hotmail.com (B.C.E.); 3Innovation and Solution Development Group, Türk Telekom Directorate, 34660 İstanbul, Türkiye; ozgur.deger@turktelekom.com.tr

**Keywords:** aged, type 2 diabetes, quality of life, teaching, self-management, HbA1c

## Abstract

Background: The purpose of the study was to determine the effect of insulin pen training using the Teach-Back method in older patients with type 2 diabetes (T2D) on their self-management of insulin treatment, quality of life (QoL), and glycated hemoglobin (HbA1c) levels. Methods: Participants included 25 patients in the intervention group, with a mean age of 80.76 ± 6.132 years, and 24 patients in the control group, with a mean age of 81.29 ± 4.920 years. All participants were older people who had previously been diagnosed with T2D, had been using insulin for at least 6 months, and lived in rural areas. Teach-Back pen training was provided to the intervention group, while general diabetes education was provided to the control group. One-way variance analysis, paired-samples *t*-test and independent sample *t*-test were used. The self-management of insulin treatment, QoL and HbA1c levels were determined before training and after 3 months. The study was conducted between December 2022 and April 2023. Results: A significant difference was found in the mean scale scores between the intervention group and control group after training. The mean self-management of insulin treatment and QoL scale scores of the intervention group were significantly higher than those of the control group after training. The post-training HbA1c levels in the intervention group were lower than the pre-training levels. Conclusions: Teach-Back training improved diabetes self-management and QoL and decreased HbA1c levels in older patients with T2D living in a rural community.

## 1. Introduction

Effective self-management of patients with type 2 diabetes (T2D) can be achieved by applying the correct technique for insulin use. The selection and use of insulin injection devices, storage conditions, selection and rotation of the injection site on the body, injection angle, correct injector needle selection, and safe waste management after use are important information that all patients should learn [1]. Using the wrong technique can cause hyperglycemic crisis or severe hypoglycemia in patients with diabetes. Hyperglycemic and hypoglycemic complications are important reasons for admission to the emergency department, and the highest rate is seen among older adults [2,3]. According to US National Surveillance data, 97,648 emergency department admissions occurred annually due to insulin-induced hypoglycemia: one-third resulted in hospitalization, and some patients had neurological sequelae [4].

Despite recent advances in insulin pen technology, errors in administration techniques remain problematic. These errors include applying insulin to the wrong body part, not using insulin pen needle tips of the appropriate length or using more than one tip, incorrectly administering the injection, and not storing the insulin pen in appropriate storage conditions. Although several factors can contribute to administration errors, a lack of training on how to operate these devices is one of the most common causes. Therefore, it may be pertinent for healthcare professionals to provide insulin pen training to all patients. Continuous assessment of the patient’s mechanical technique is also required to reinforce training when necessary [5].

The Teach-Back method involves asking patients to explain or demonstrate what they have been told. It is a method that can be easily used in all kinds of interactions between the healthcare team and patients [6]. This method is important for increasing compliance with treatment plans and ensuring patient safety [7]. Moreover, this approach can be used to assess a patient’s technique, reduce practice errors that can lead to complications and hospitalizations, and retrain them whenever possible [5].

The Teach-Back technique is successfully used in the care management of many chronic diseases such as heart disease and asthma [6,7]. It helps strengthen communication between patients and healthcare professionals and is associated with a decreased risk of disease complications and re-hospitalization [8].

Recently, this technique has begun to be used in the training given to patients with diabetes; however, the literature on this subject is extremely limited.

In a control group study conducted on a sample of all age groups with T2D diagnosis and low literacy levels in a diabetes clinic, the Teach-Back method and pictorial image methods were used for the intervention group. The groups that received education with the pictorial image and Teach-Back methods had higher diabetes knowledge, medication and diet compliance scores than the control group, which received standard diabetes education [9].

The role of patient and healthcare provider interactions in diabetes care delivery was examined based on the data from the Medical Expenditure Panel Survey 2011–2016 study conducted in the USA. The study included individuals with diabetes aged 18 and over. Patients who used the Teach-Back method had higher scores in terms of the quality of interaction with their healthcare providers. These patients were more likely to receive additional advice from healthcare providers about diet and exercise and reported that they were confident in their diabetes self-care management. As a result of this study conducted with a large sample of 2900 people, it was reported that the Teach-Back method was very effective in diabetes education but was not used enough and should be included in routine diabetes care models [10].

In another study using the Teach-Back method, patients with T2D were given physical activity, foot care and glycemic control training. The Teach-Back group had higher diabetes self-efficacy scores than the group that watched a videotape on the same topics [11].

None of the studies mentioned were aimed at older adults, included people living in rural areas, or were conducted in the homes of patients in the community.

The Teach-Back method was also used to improve the health literacy of health ambassadors. In a study conducted in a district, the health ambassadors were individuals aged 14 and over. In this study, which was a project in which one person from each family was selected as a health ambassador to inform their families about health, the health literacy scores were higher in the group using the Teach-Back method [12].

When other studies conducted on diabetic patients using the Teach-Back method were examined, we found that all of the studies were conducted inside the hospital, were conducted for patients who started using insulin for the first time, and included participants of all ages over the age of 18 years [1,13,14]. To the best of our knowledge, no study on the Teach-Back insulin pen training method in older people with T2D living in rural communities has been conducted. The education level and health literacy levels of older people living in rural areas are lower. Access to healthcare and the utilization of public social services are much more difficult for older people living in rural areas.

Therefore, this study aimed to determine the effect of insulin pen training using the Teach-Back method to older patients with T2D living in rural communities on self-management of insulin treatment, quality of life (QoL), and glycated hemoglobin (HbA1c) levels. The current study sample consisted of participants who were older adults, had T2D, and lived in rural areas; hence, they had three disadvantages.

## 2. Materials and Methods

This was a quasi-experimental study with a pretest–posttest design focused on older people with T2D aged 65 years and older living in a community in Eldivan, Çankırı, Türkiye.

### 2.1. Study Setting

The study was conducted in a small town with a total population of 3185 people. As job opportunities were limited, young people migrated to large cities and the proportion of older people living in the town increased (17.08%). When calculated based on this ratio, the number of older people living in the town was approximately 544 people [15]. There was only a small primary health center in Eldivan. In areas requiring specialization (e.g., internal medicine), older people had to go to the city. Therefore, access to health services was limited. The majority of the older people were women, did not have their own income, lived on their husbands’ salaries, and had low levels of education (45.8% were illiterate in the current study). This study was conducted among older adults with T2D living in the community and in their homes. Health center records from the town were used to identify older adults with T2D, and interviews were conducted by visiting their homes. Older adults with T2D who voluntarily agreed to participate and met the study criteria were included. The training at all stages of the study was conducted in the homes of the study participants. Thus, health services were brought to the participants. The special needs of a disadvantaged group, characterized by advanced age, low education level, having T2D and living in rural areas, were also met. In addition to the academic aspect of the study that contributed to science, there was also an aspect that contributed to society.

### 2.2. Study Sampling

This study included intervention and control groups. A similar previous study should be referenced to calculate the sample size. In a study conducted by Jing et al., patients who started using insulin for the first time were made to watch a video using the Teach-Back method, and a control group study was conducted [1]. The minimum sample size for the current study was calculated based on the effect size value of this study, with power of at least 80%. Accordingly, the minimum sample size was 46 (23 participants in the intervention group; 23 participants in the control group), with an effect size of 1.08, alpha value of 0.05, and power of 0.80. To account for losses during the 3 months, 10% more than the sample size was included in the study; hence, a total of 50 older adults with diabetes were included.

The inclusion criteria included those who were aged 65 years and older, living in their own homes in Eldivan, diagnosed with type 2 diabetes mellitus at least 6 months ago, using insulin as a treatment for diabetes, and agreed to participate in the study.

The exclusion criteria included those who were unable to inject insulin or were injected by caregivers, who had hearing and vision problems, were receiving treatment other than insulin as a diabetes treatment, and those without a diagnosis of a psychiatric or cognitive disease (dementia, etc.).

### 2.3. Determination of Groups

Three stratified methods were used. Participants were grouped equally according to age, educational level, and health literacy level. The first stratification was based on age group. Participants were stratified based on three different age groups: 65–74 years, 75–84 years, and 85 years and above. The second stratification was based on education level and the third stratification was based on health literacy scale scores. Since the study was conducted in the field, it was very difficult to create a control group, so a quasi-experimental method was used. The groups were matched as closely as possible in terms of age, education level, and health literacy levels.

### 2.4. Study Design

The research team consisted of a gerontologist and two nurse researchers (one a diabetes nurse and the other a statistician), all of whom were also the authors of the study. Before creating the intervention and control groups, scales were administered to all participants. Among these scales, the Health Literacy Scale (HLS-14) was not used to evaluate the effect of the study on the participants but only to contribute to the homogeneity of the intervention and control groups. The principle of complete confidentiality was applied while creating the intervention and control groups. None of the participants were given the names or information of the other participants. The training step was then initiated. Training was provided face-to-face in the participants’ homes by a diabetes nurse and a nurse researcher and was conducted once a week for a total of 4 weeks.

### 2.5. Training Implementation in the Intervention Group

The Teach-Back insulin pen training method was applied to the intervention group. In this method, after older patients with T2D were provided information and taught, they were asked to repeat and explain it in their own words. Open-ended questions were asked by an instructor, who was a diabetes nurse, to evaluate the patient’s knowledge-learning status. The instructor evaluated the degree to which information was learned from the patients’ answers to the questions. Additional information was provided when the patients’ explanations were deemed sufficient. Moreover, patients demonstrated the technique they had learned to the instructor by reapplying and explaining it. The training was finalized by following the steps of teaching, teaching back, and evaluation. The instructor used simple language to avoid medical terminology [13].

All participants in the intervention group were older adults with T2D who had started using insulin at least 6 months ago. When insulin was first prescribed in the hospital months or years ago, the patients were shown how to use it by doctors, nurses, or pharmacists. The patient was not educated again thereafter. In the current study, with the Teach-Back insulin pen training provided by diabetes nurses in the homes of older patients, the patients were able to see and correct their faulty practices. The intervention group consisted of 25 older individuals with T2D.

Teach-Back insulin pen training was given to participants in their homes once a week for 4 weeks. Each training session included 30 min of Teach-Back insulin pen training and 15 min of standard diabetes education for a total of 45 min.

The titles of the insulin pen training sessions were as follows:

Week 1: Insulin storage, carrying methods, selection, and management of devices.

Week 2: Insulin application areas and steps, selection of the insulin injection zone, rotation sequence of repeated doses, correct angle application, and correct injector needle length selection.

Week 3: Complications of insulin therapy and prevention.

Week 4: Proper waste management of post-use devices and injector needles [1].

### 2.6. Training Implementation in the Control Group

The control group also received diabetes education; however, the Teach-Back method was not used. Participants were visited in their homes once a week for 4 weeks, and standard diabetes education was provided; however, the patient was not asked to explain the method again or demonstrate it by applying it. Standard diabetes education included information about self-monitoring of blood glucose levels, prevention and management of complications of diabetes mellitus (e.g., prevention methods for diabetic foot), strategies of nutrition in diabetes, and living with diabetes. One of the participants in the control group could not accept the educator in her home for a week due to an acute illness; therefore, she was excluded from the study, and the control group included 24 participants.

### 2.7. Instruments

All participants completed the Patient Information Form, HLS-14, Insulin Treatment Self-Management Scale (ITSMS), and Older People’s Quality of Life-Brief Scale (OPQOL-Brief). HbA1c blood values were measured in all participants. The scales used are not under any license. In addition, scale use permissions were obtained via e-mail from the authors who conducted and published the validity and reliability studies of the scales on the local population before the ethics committee application. The HLS-14 was administered only once before the study. All other scales were administered before and 3 months after training. Blood HbA1c values were also measured before training and 3 months after the training ended. After all preparations were completed, the first pre-training scale was applied in December 2022. The training was conducted in January 2023, and the final scales were applied in April 2023. Blood HbA1c levels were also measured on the same dates.

#### 2.7.1. Patient Information Form

The patient information form was a 13-item questionnaire that included information on the age, sex, and current diseases in the participants. This form was developed by the researchers.

#### 2.7.2. HLS-14

The HLS-14 scale was used to create homogeneous intervention and control groups before the study. Intervention and control groups with similar ages and education levels were formed according to similar health literacy scores. Thus, the intervention and control groups were more homogeneous, and the academic results of the study were stronger.

A validity and reliability study of the scale was conducted among local older adults. The scale has three sub-dimensions: functional (five items), interactive (five items), and critical (four items). An increase in the total score indicated an increase in health literacy [16].

#### 2.7.3. ITSMS

The validity and reliability of the ITSMS scale in the local population has been assessed, and it has been reported to be a valid and reliable scale for patients with diabetes. Cronbach’s alpha was 0.91 [17].

#### 2.7.4. OPQOL-Brief

OPQOL-Brief was used to measure the patients’ QoL. This scale was developed by Bowling et al. to assess the QoL in the geriatric population [18]. A validity and reliability study of the OPQOL-Brief scale was conducted for local older people. The OPQOL-Brief scale, consisting of 13 items, is answered as a likert type scale of 1 to 5, where 1 indicates “strongly disagree” and 5 indicates “strongly agree” for each item. The total score ranges from 13 to 65. Higher scores indicate a better QoL [19].

#### 2.7.5. HbA1c Level

Hba1c levels were obtained from records measured at the health center in the town where the patients were enrolled. Patients were called to the health center before the study and 3 months after the training, and blood Hba1c levels were measured with the approval of the patient’s family physician and through the patient’s health insurance.

### 2.8. Data Analysis

The data obtained in this study were analyzed using Statistical Package for Social Sciences for Windows version 25.0 (Armonk, NY, USA: IBM Corp.). To determine the similarity between the control and intervention groups in conditions that may affect the educational outcome, such as age, duration of diabetes, health literacy scores, gender distributions, education levels, phone use situations and living alone situations, a *t*-test and one-way analysis of variance for the difference between the averages of two independent groups (ANOVA) was used.

The paired-sample *t*-test was used for normally distributed data in the pretest and posttest comparisons. An independent samples *t*-test was used to compare the control and intervention groups for normally distributed data.

### 2.9. Ethical Considerations

This study was approved by the University Ethics Committee before starting the study (decision number: 23; decision date: 9 November 2021). This study was conducted in accordance with the Declaration of Helsinki and other international ethical guidelines. Informed consent was obtained from all participants.

## 3. Results

The mean age of the control group was 81.29 ± 4.920 years, and the mean duration of diabetes was 23.41 ± 2.500 years. In the intervention group, the mean age was 80.76 ± 6.132 years, and the mean duration of diabetes was 23.84 ± 2.718 years. There were no statistically significant differences between the sociodemographic characteristics of the two groups (*p* > 0.05) (Table 1). It was observed that there was no difference between the control and intervention groups in terms of conditions that could affect the educational outcome such as age, duration of diabetes, health literacy scores, gender distribution, educational status, phone usage status and living alone (*p* > 0.05). The intervention and control groups consisted of similar groups in terms of some characteristics (Table 1).

Regarding disease characteristics, 83.3% of the intervention group had an additional chronic disease, 79.2% did not use oral antidiabetics, 79.2% used insulin four times a day, 37.5% received information on insulin use from a doctor, 91.7% had difficulty using insulin, and 45.45% had forgotten how to use an insulin pen. There were no statistically significant differences in disease characteristics between the two groups (*p* > 0.05) (Table 2).

When the ITSMS was examined, the average scores after training were significantly higher in both groups than before training (control group: *p* = 0.045; intervention group: *p* = 0.000). The post-training average scores of the intervention group were significantly higher than those of the control group (*p* = 0.000). The behavioral, cognitive, and affective subscale scores and their significance are presented in Table 3.

When the OPQOL-Brief was analyzed, a statistically significant difference was found in the mean scores of the total scale of the life quality scale in the participants before and after training in the control group. The mean scores after training were significantly higher than those before training (*p* = 0.001). In the intervention group, a statistically significant difference was found in the mean scores of the total scale of the life quality scale before and after training. The mean scores after training were significantly higher than those before training (*p* = 0.000). Although there was a statistically significant difference in the mean scores of the total scale of the life quality scale between the intervention and control groups, the mean scores of the intervention group after training were significantly higher than those of the control group (*p* = 0.000) (Table 3).

## 4. Discussion

This study had three outcomes: diabetes self-management, QoL, and HbA1c level.

### 4.1. Diabetes Self-Management

When the literature in which the Teach-Back method was used and its effect on diabetes self-management were examined, no studies were found that included older people living in the community. Studies conducted in hospitals including all age groups were examined. In a study in which patients with T2D in the endocrinology clinic were trained using the Teach-Back method, self-efficacy scale scores were significantly higher in the intervention group compared to the control group 1 month after the education were found [20]. In this study, Farahaninia et al. showed that Teach-Back significantly increased patients’ self-efficacy even in a short period of 1 month and reported that it might be interesting to examine the long-term effects of this simple but effective training method.

The current study showed an increase in diabetes self-management 3 months after the end of the training exercise, thus taking the research topic further, as suggested by Farahaninia et al.

In another study using Teach-Back insulin training, patients diagnosed with and using insulin for the first time were trained using videos and paper documents. Insulin management skills were measured with a scale called “My View on Insulin”, and the scale scores increased after training [1].

In a study conducted at a diabetes clinic in Iran, participants’ self-care performance was measured. Only educated people were included in the study, and the average participant age was 55 years. In a study consisting of a Teach-Back training group, an online training group, and a control group, the Teach-Back training group showed the best performance according to an independent *t*-test [14].

The results of the current study are consistent with the aforementioned studies in terms of diabetes self-management outcomes. In contrast, training was conducted face-to-face for older people living in rural areas. Moreover, as in the sample selected by Hemmatipour et al., an increase in diabetes self-management was achieved not only in those with a certain level of education but also in those who were illiterate. In the intervention group, a statistically significant increase was observed in all sub-scores of the self-management scale in the third month after training compared with before training. Although approximately half of the sample in the study consisted of illiterate elderly people, the effectiveness of the teaching method, in which older adults first listened to and then explained themselves, was strikingly observed.

### 4.2. Quality of Life

In a study with a control group comprising 74 patients with T2D in the Endocrine and Metabolism Clinic of a hospital, a four-session training program was administered to the participants in the intervention group using the Teach-Back method. The control group underwent routine programs only. One month after the completion of the training sessions, the Lifestyle Profile survey results were more significant in the intervention group than in the control group. The study was conducted for a period of 1 month, and it is recommended that the effect of this method be investigated for longer periods of time in order to evaluate its long-term effects [21].

In the current study, an increase in the QoL was found after training, which is consistent with the literature. The group that received Teach-Back insulin education had higher QoL scores than the group that received standard diabetes education. It was thought that the QoL of patients with T2D increased owing to their acceptance of diabetes, self-management of diabetes, ability to cope with the complications of diabetes, and the education they received. Adding Teach-Back insulin pen training further improved their QoL. For this reason, supporting patients with T2D with education, using the Teach-Back technique, and continuing this education not only in the hospital or when the diagnosis is first made but also while living in the community may play an important role in increasing patient QoL.

Other studies were searched in the literature to show the effect of Teach-Back insulin education on the QoL of T2D patients, but none were found. The current study is unique in this respect. Although not directly related, some studies were found. In a hospital, the QoL was found to be higher in the intervention group in which Teach-Back diet management education was given to diabetic patients with hepatitis B than in a control group in which traditional diet management was applied [22]. In another study, a 7-day inpatient diabetes education program was applied to hospitalized patients with T2D, and their QoL increased [23]. A cross-sectional study found that as the diabetes burden scale scores increased in the older adults, their QoL decreased [24]. A systematic review including randomized controlled trials was examined. In three of the nine trials in which the structured diabetes education program was applied, the QoL was found to be higher in the intervention group than in the control group [25].

In the current study, the QoL scores of the participants increased in both the control group in which routine diabetes education was given and in the group in which Teach-Back education was given. In addition, the QoL score was significantly higher in the intervention group compared to in the control group 3 months after the education. Thus, the importance of Teach-Back insulin pen training in the QoL of patients with T2D was clearly demonstrated.

### 4.3. HbA1c Level

A limited number of studies investigating the effects of the Teach-Back method on HbA1c levels were examined. In one study, patients who had been on insulin therapy for at least 3 years were asked questions about their insulin injection technique, their level of knowledge was scored, and they were then subjected to training by physicians. A decrease in the HbA1c levels of the patients was found 2, 3, and 4 months after the training [26]. In the endocrinology clinic of a medical faculty hospital, patients with T2D, whose average age was 54 years and who started insulin treatment for the first time, were divided into intervention and control groups of 35 patients each. Three months after the training, participants’ diabetes knowledge levels increased and HbA1C values decreased [13]. Similarly, in the current study, there was a decrease in HbA1c levels 3 months after training.

Since the current study was conducted on older people living in a rural community, similar studies were analyzed. In one study, patients with T2D living in a rural town and going to the health center of that town for check-ups every 3 months were trained using the Teach-Back method. Twelve patients with T2D were included in the study, and there was no control group. Participants’ HbA1c levels decreased after 3 months [27].

Hence, we believe that the decrease in HbA1c levels is due to the use of the correct technique thanks to the Teach-Back method and an increase in patients’ compliance with diabetes management and treatment.

In addition to studies showing that the Teach-Back technique is effective, there are studies showing that it is not useful. In one study, teaching techniques were added after the implementation of a multimedia diabetes education program. If the participants answered the question incorrectly after watching the video, a model was followed in which the information was reviewed, and the question was asked again at least twice. In the evaluation made after 15 days, it was reported that the participants had forgotten half of the information [28]. It was understood that the application technique and method of teaching back also played a significant role in success.

When the current study was evaluated at an internal level, it was conducted in rural areas. Only patients with T2D aged 65 years or older were included in the sample, and approximately half of them were illiterate. The current study positively impacted the lives of older people included in the sample. It also increased the awareness among older people living in towns about diabetes self-management. Moreover, regarding social policies, it has been shown that it is important to make home visits to older adults living in rural areas, and that preventive and rehabilitative health service provision yields beneficial results.

### 4.4. Limitations

The first limitation of this study was the difficulty in convincing older adults, who were not used to such studies and initially could not comprehend the benefits of participation, to take part. Therefore, a randomized study could not be performed because it was impossible to convince the participants to participate in a prospectively registered study. Although the study was not randomized, when creating the intervention and control groups, an attempt was made to create groups as equal as possible in terms of the age ranges, education levels and health literacy levels of the participants. This is also why a health literacy scale was applied to the participants before the study. In this way, an effort was made to ensure that the study results were more reliable.

The second limitation of this study is that the findings, which are specific to rural older adults, cannot be generalized to urban older people. The findings are specific to the culture in which they were conducted and cannot be generalized for other nations and cultures. Multicenter and registered randomized studies are needed for future studies.

## 5. Conclusions

In this study, patients with T2D were trained using the Teach-Back method. After training, a significant increase in the mean total scores of the intervention group’s ITSMS and life quality scale compared with the control group was found. Moreover, a significant decrease in HbA1c levels was also noted in the intervention group. Hence, Teach-Back training improved diabetes self-management and life quality and decreased HbA1c levels in an older cohort of patients with T2D living in a rural community.

The Teach-Back education techniques that have just begun to be implemented in recent years should be disseminated. Every patient who starts using insulin should receive Teach-Back training at the health center, which should be repeated at routine intervals. Training in this technique should be provided to older people with T2D, especially within the services of home care teams for older adults. Teach-Back insulin pen training should be included in national health policies and should be disseminated by home health teams. When providing this service, priority should be given to patients who live in rural areas, have limited access to well-equipped health centers, and are older adults.

This study may create momentum for further research. Pilot studies of Teach-Back diabetes training within home care teams in rural and urban areas could be conducted. New home care models may be designed from long-term results.

## Figures and Tables

**Table 1 healthcare-12-01854-t001:** Comparison of participant sociodemographic characteristics according to intervention group.

Variables		Control		Intervention		Test Value/*p*
		N	%	N	%	
Age (years)		81.29 ± 4.920		80.76 ± 6.132		*t*: 0.334 ** *p*: 0.740
Diabetes duration (years) (mean ± SD)		23.41 ± 2.500		23.84 ± 2.718		*t*: −0.567 ** *p*: 0.574
Health literacy score		26.54 ± 1.284		25.88 ± 1.423		*t*: 1.706 ** *p*: 0.095
Age (years)	65–74	2	8.3%	2	8.0%	F: 0.025 * *p*: 0.984
	75–84	13	54.2%	13	52.0%	
	85 and above	9	37.5%	10	40.0%	
Sex	Woman	19	79.2%	20	80.0%	*t*: 0.071 ** *p*: 0.942
	Man	5	20.8%	5	20.0%	
Marital status	Married	19	79.2%	20	80.0%	*t*: 0.071 ** *p*: 0.942
	Loss of spouse	5	20.8%	5	20.0%	
Who do you live with?	With spouse	14	58.3%	15	60.0%	F: 0.011 * *p*: 0.993
	With children	5	20.8%	5	20.0%	
	Alone	5	20.8%	5	20.0%	
Education status	Illiterate	11	45.8%	11	44.0%	F: 0.004 * *p*: 0.947
	Literate	10	41.7%	11	44.0%	
	Primary School	3	12.5%	3	12.0%	
Telephone use	Home phone	8	33.3%	8	32.0%	F: 0.007 * *p*: 0.932
	Cell phone with internet	6	25.0%	6	24.0%	
	Cell phone without internet	5	20.8%	6	24.0%	
	No phone	5	20.8%	5	20.8%	

* One-way variance analysis (ANOVA); ** *t*-test in independent groups; SD, standard deviation.

**Table 2 healthcare-12-01854-t002:** Comparison of participant disease characteristics according to intervention group.

Variables		Control		Intervention		Test Value/*p*
		N	%	N	%	
Presence of comorbid chronic diseases	Yes	20	83.3%	21	84.0%	*t*: −0.062 ** *p*: 0.951
	No	4	16.7%	4	16.0%	
Use of oral antidiabetics in addition to insulin	Yes	5	20.8%	5	20.0%	*t*: 0.071 ** *p*: 0.942
	No	19	79.2%	20	80.0%	
How many times a day do you administer insulin?	Twice a day	5	20.8%	5	20.0%	*t*: 0.071 ** *p*: 0.942
	Four times a day	19	79.2%	20	80.0%	
Who trained you for insulin pen use?	Nurse	6	25.0%	7	28.0%	F: 0.141 * *p*: 0.709
	Doctor	9	37.5%	10	40.0%	
	Pharmacy	9	37.5%	8	32.0%	
Difficulty in using insulin	Yes	22	91.7%	21	84.0%	*t*: −0.807 ** *p*: 0.424
	No	2	8.3%	4	16.0%	
While using an insulin pen	My hands are shaking	5	22.7%	4	19.04%	F: 0.651 * *p*: 0.424
	Difficult to adjust the dosage	7	31.8%	6	28.57%	
	I forget it	10	45.45%	11	52.38%	

* One-way variance analysis (ANOVA); ** *t*-test in independent groups.

**Table 3 healthcare-12-01854-t003:** Comparison of mean scores by groups.

Scales		Control Mean ± SD	Intervention Mean ± SD	*** *t*-Test	*p*
Insulin Treatment Self-Management Scale					
Behavioral sub-dimension (17–85 points)	Before	62.29 ± 3.071	62.64 ± 3.289	−0.383	0.705
	After	62.41 ± 3.133	70.32 ± 3.771	−7.313	**0.000 ***
	*t*-test **	4.033	17.509		
	*p*	**0.001 ***	**0.000 ***		
Cognitive sub-dimension (7–35 points)	Before	21.79 ± 2.587	22.48 ± 2.583	−0.932	0.356
	After	22.83 ± 2.277	29.04 ± 1.019	−12.395	**0.000 ***
	*t*-test **	−3.824	−5.4480		
	*p*	**0.001 ***	**0.000 ***		
Affective sub-dimension (8–40 points)	Before	30.75 ± 2.523	30.64 ± 2.612	0.150	0.882
	After	30.25 ± 2.996	35.12 ± 2.773	−5.907	**0.000 ***
	*t*-test **	0.796	−6.965		
	*p*	0.434	**0.000 ***		
Insulin Treatment Self-Management Scale total score	Before	114.83 ± 4.555	115.76 ± 4.474	0.718	0.476
	After	116.45 ± 4.916	134.48 ± 4.874	−12.883	**0.000 ***
	*t*-test **	−2.122	−22.035		
	*p*	**0.045 ***	**0.000 ***		
Life Quality Scale for the Elderly Short Form					
Total score	Before	44.45 ± 4.043	45.04 ± 3.724	−0.524	0.603
	After	44.79 ± 3.977	47.60 ± 4.320	−2.364	**0.022 ***
	*t*-test **	−2.145	−5.818		
	*p*	**0.043 ***	**0.000 ***		
HbA1c Value					
HbA1c (mg/dL)	Before	10.77 ± 0.728	10.8880 ± 0.64117	−0.556	0.811
	After	10.72 ± 0.661	10.4600 ± 0.673	1.411	0.165
	*t*-test **	1.468	12.617		
	*p*	0.156	**0.000 ***		

* *p* < 0.05; ** paired-samples *t*-test; *** *t*: independent sample *t*-test; HbA1c, glycated hemoglobin; SD, standard deviation. *p* values of statistically significant results are written in bold.

## Data Availability

The data presented in this study are available upon request from the corresponding author as they pertain to the older adults in the rural district of Eldivan and to respect their privacy.

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
