# Peer review of "Effect of Insulin Pen Training Using the Teach-Back Method on Diabetes Self-Management, Quality of Life, and HbA1c Levels in Older Patients with Type 2 Diabetes: A Quasi-Experimental Study"

_healthcare, 2024, doi:10.3390/healthcare12181854_

Round 1

Reviewer 1 Report

Comments and Suggestions for Authors

Dear Authos,

file annex.

Good work

Reviewer 2 Report

Comments and Suggestions for Authors

The authors present a theme of actuality that is characterized by importance and strong medical interest. 

The manuscript is well written and I consider that all main points have been debated. 

The suggestions I have is that:

  • English spelling should be revised.
  • In the abstract, the age for the participants and some inclusion criterias, the period of time the study was performed in material and method and some specific results (data analysis perhaps) should be introduced 
  • Row 42: please give some examples of errors for administration
  • I suggest that in the introduction section to add more datas or conclusions supported by the medical literature 
  • The explanation for the Teach Back method (rows 47-49), I consider it fits better to material and methods section
  • Row 84 : please detail the study of Jing et al. 
  • Row 89: did the patients sign an informed consent? Please add
  • Please explain if the scales and questionnaires provided were under any licence or they were free to be applied for the patients
  • please explain the importance of the results obtained in the results section from the data analysis
  • Also please add some data analysis from the studies presented in comparison to your study in the discussion section. 

Thank you.

Comments on the Quality of English Language

 Minor editing of English language required.

Round 2

Reviewer 1 Report

Comments and Suggestions for Authors

Dear Authors,

the comments in file annex.

Best

Reviewer 2 Report

Comments and Suggestions for Authors

All the requests have been completed have been completed one way or another.

Comments on the Quality of English Language

Moderate editing of English language required but the authors declared that they would request MDPI english services.